# Application of Fuzzy Logic for Problems of Evaluating States of a Computing System

**Abror Buriboev [1] , Hyun Kyu Kang [1], Myeong-Cheol Ko [1], Ryumduck Oh [2], Akmal Abduvaitov [3] and Heung Seok Jeon [1,\*]**

[1]   Department of Software Technology, Konkuk University, Chungju 27478, Korea
[2]   Department of Software, Korea National University of Transportation, Chungju 27469, Korea
[3]   Department of Information Technologies, Tashkent University of Information Technologies, Samarkand 140100, Uzbekistan
[\*]   Correspondence: hsjeon@kku.ac.kr; Tel.: +82-43-840-3621

**Abstract:** The monitoring utilization and workloads of computer hardware components, such as CPU, RAM, bus, and storage, are an ideal way to evaluate the effectiveness of these components. In this paper, we surveyed the basic concepts, characteristics, and parameters of computer systems that determine system performance, and the types of models that provide adequate modeling of these systems. We investigated and developed the applied aspects of the theory of fuzzy sets' principles and the Matlab environment tools for monitoring and evaluating the state of computing systems. The idea of the paper is to identify the state of the computer infrastructure by using the models of Mamdani and Sugeno FIS (fuzzy inference system) to evaluate the impact of RAM and storage on CPU performance. With this approach, we observed the behavior of computer infrastructure. The results are useful for understanding performance issues with regard to specific bottlenecks and determining the correlation of performance counters. Moreover, the model presents linguistic results. Hereafter, performance counter correlations will support the development of algorithms that can detect whether the performance of a given computer will be affected by a reasonable priority. The performance assertions derived from these approaches allow resource management policies to prevent performance degradation, and as a result, the infrastructure will be able to serve safely as expected. These methods can be applied across the entire spectrum of computer systems, from personal computers to large mainframes and supercomputers, including both centralized and distributed systems. We look forward to their continued use, as well as their improvement when it is necessary to evaluate future systems.

**Keywords:** complex performance; fuzzy logic; membership functions; Mamdani and Sugeno fuzzy inference system

## 1. Introduction

Computer performance depends on how well the hardware components work together and interact as a whole system. Continuously updating one part of the computer while preserving obsolete components will not lead to a significant increase in computer performance, if at all. Hereafter, we survey some of the most important hardware components of a computing system regarding its efficiency and computing power. The description of these parts does not mean complete definition and serves only to give new users some idea of what the diver characteristics of computer performance mean. The central processing unit (CPU), the RAM, the bus, and the storage are the most important components in determining performance inside a computer. The degree of compliance of the system with its purpose is called the efficiency (quality) of the system. For complex systems like computing

systems, efficiency cannot be determined by one quantity, and therefore, it is represented by a set of quantities called system characteristics. The set of characteristics is formed in such a way that, on aggregate, they give the most complete picture of the effectiveness of the system [1]. The main characteristic of a computing system is performance. The tasks of evaluating the effectiveness of the organization of computing systems are defined as mathematical objects, called system parameters. As parameters, values are used that determine the number and speed of devices, memory capacity, workload, etc., in fact, all objects that characterize the primary aspects of the organization of the system and significantly affect the characteristics.

Performance is a characteristic of the computing power of a system that determines the amount of computing work performed by the system per unit of time. Currently, there is no generally accepted method for assessing the performance of a computing system, which is primarily due to the lack of units for measuring the amount of computational work. Therefore, to evaluate the performance, a wide range of values are used—performance indicators—that both individually and in aggregate do not fully meet the needs of the theory and practice of designing and operating a computing system [1].

Nominal performance characterizes only the potential capabilities of devices that cannot be fully utilized. This is hampered by the influence of the structure of communications between devices on their performance, which is manifested in a change in the speed of operation of some devices when others work. Therefore, due to the fact that the processor and I/O channels are connected to common RAM, an increase in I/O speed leads to an increase in processor performance; the total performance of I/O devices connected to the multiplex channel is limited by the channel bandwidth, the actual performance of the drives connected to the block-multiplex channel is less than their total nominal speed, etc. [2,3]. To evaluate the influence of the first group of factors—the system structure on the performance of devices—a special characteristic is used: complex performance. Complex performance is estimated by the set of speeds of the devices provided by their joint work. There are a lot of ways to assess integrated performance. One approach to its assessment is as follows. In some way, a typical mixture of input operations, access to external memory, processing and output of data is determined, based on which, a synthetic artificial program is created that generates a process with a given mixture of operations. By running a synthetic program and measuring its runtime, the integrated system performance is estimated [4].

The influence between the systems is manifested, for example, in the following. The organization of system input and output is associated with the use of a processor and external storage devices for intermediate storage of input and output datasets. As a result, part of the processor time, I/O channels, and external storage devices are spent on I/O service. The same situation arises when organizing a virtual memory system, time-sharing mode and providing other auxiliary functions. For computing systems that are in operation or are being developed for a specific application, the class of tasks is fully defined, at least statistically. That is, the workload of the computing system is determined. In this case, performance is evaluated on the workload and is called system performance [5]. It is mainly the number and speed of devices, the capacity of RAM and external memory (with an increase in performance if these factors increase), as well as the structure of the system and the bandwidth of connections between the elements of the system that have the most significant effect on performance. The performance of a computing system is manifested, on the one hand, in the speed of processing tasks, and on the other, in the degree of use of system resources. The more resources are loaded, the higher the system performance, and underloading of resources indicates the presence of reserves to improve performance. Therefore, when analyzing system performance, not only are performance indicators evaluated, but also indicators characterizing the use of resources.

## 1.1. Motivation

There are three main objectives to evaluate computer states: computer selection, computer designing, and configuration enhancement [6]. In all these fields, the goal is to optimize the response of the actual or projected system to the actual or predicted workload. Assessment of the state of computers

is attempting to define how well a particular system corresponds or can satisfy certain processing requirements for data or planned resources. There are several available assessment methods, the bulk of which are appropriate to some of the objectives, for instance, synthetic programs and benchmarks are considered the best evaluation methods. However, these evaluation techniques are not based on an analysis of real data taken from the existing real systems. All fuzzy based CPU utilization predictive models are based on tendency and previous workload states [7] or consider historical data of CPU (cores, response time), memory (reading, writing, swapping time, queue), bus (I/O throughput, latency), etc., that are too long. Moreover, each object has a variety of pointers (time, workload, frequency, and percent) that complicates the evaluation task. However, all of the above methods are complicated despite a good result. Many of them require data that are difficult to obtain for each user and require special programming skills, such as machine learning algorithms, reference datasets for evaluating performance and in-depth knowledge of computer architecture. Moreover, the results of evaluating the performance or resource utilization presented by the approaches discussed above are not fully understandable to the user. Based on the explanations above, it is necessary to simplify and convert all data types to unequal metrics, i.e., linguistic variables using fuzzy set theory. This paper aims to build an evaluation model for any type of computer systems from personal computers to large mainframes and supercomputers that can be used by non-professional users who have no analytical knowledge about hardware components. The proposed model should evaluate the state of the computer system, CPU utilization, determine the incompatibility and bottleneck of hardware components and predict the CPU utilization in given condition cases.

### 1.2. Related Work

The related work in this field has been done by different simulations, environment and performance tools. In various studies, analysts have considered the following factors while evaluating the computer performance: waiting time, load balancing, number of requests, the throughput and rate of transactions. These computer performance factors were evaluated by many approaches. We classified them as fuzzy set theory based and machine learning (prediction) based approaches.

#### 1.2.1. Fuzzy Set Theory Based Approaches

Butt et al. [8] have developed a fuzzy decision-making system to improve the CPU scheduling algorithm in a multitasking operating system. They used a new formula to calculate the recent CPU utilization by each process. In addition, the algorithm maintains time intervals and recalculates the dynamic priority of the processes upon the arrival of the high-priority process and after the allotted time of the process. They have developed the simulator so that it takes the packet time, the finished value and the arrival time of each finished process as input. Then, the simulator calculates the recent CPU utilization of each process using the formula. A FIS generates a fuzzy dynamic priority (dpi) for each process. In accordance with the dpi of each process in descending order, the process at the head of the queue is selected to run on the processor. The proposed performance evaluation method by Jung et al. [9] measures the online computer systems using the modeling occurrences of failed computer hardware units based on fuzzy set theory. In this method, it is mandatory to use subjective possibilities and probabilities when there is not enough information about some parameters of a model. Information can be obtained subjectively from experts or very little data. They present the fuzzification model and performance evaluation of online computing systems with failure based on fuzzy set theory [9]. Suzer et al. [10] proposed and developed a fuzzy controller for utilization management. Their developed fuzzy controller can easily adapt its control actions, i.e., rules that consider the real-time system behavior. This means the controller could manage the system workload.

A self-adaptive algorithm for predicting the cloud resources using a fuzzy neural network has been proposed by Chen et al. [11]. They used the combined algorithms of Fuzzy C-means and the subtractive fuzzy clustering algorithm to optimize the characteristics of convergence and learning speed. Their fuzzy neural network learning algorithm is optimized using a self-adjusting learning

rate and impulse weight, which improves reliability and performance in real time. Beghdad et al. [7] proposed an approach aiming to develop a model one step before CPU utilization prediction based on the clusterized selection of instances in the past steps. Proposed adaptive network-based fuzzy inference (ANFIS) predicts using the Naive Bayesian Network controller. They used the C-means clustering method, which estimates the next step direction in the time series. The second proposed approach by Beghdad et al. [12] shows the future of CPU usage prediction based on a mixture of ANFIS models to compute short-term accuracy and mid-term reliable prediction. Sh. Javad [13] presented an attempt to apply neuro-fuzzy in the development and realization of a rule-based planning algorithm to eliminate the disadvantage of known planning algorithms. The decision maker on the basis of fuzzy data was asked to calculate the new priority of all CPU processes in accordance with the priority of the process and the time of its execution.

### 1.2.2. Machine Leaning Based Approaches

Mason et al. [14] proposed an evolutionary neural network to predict host CPU utilization using several different recurrent neural networks. The authors used a dataset that used PlanetLab files and three optimization algorithms to optimize weights for RNN, namely: optimization of a particle swarm (PSO), differential evolution (DE) and evolutionary adaptation strategy of the covariance matrix (CMA-ES). Duggan et al. [15] used a machine learning based approach for forecasting CPU utilization. They utilized a recurrent neural network trained using the algorithm known as backpropagation over time to predict host CPU utilization in the Google Cluster trace dataset. Kumar et al. [16] developed a novel model for forecasting the workload of cloud data centers using LSTM networks. They tested the model using three benchmark datasets of HTTP traces of the NASA server, Saskatchewan server, and Calgary server. The proposed model achieved significant results. Their proposed model can perform a 60-min prediction of the workload of big datacenters.

Two methods for predicting the resource consumption were presented by Tan et al. [17]. The first method predicts the resource usage for a specific node and uses both CPU and memory usage for its performance predictions. The second method focuses on a large number of nodes and the workload running on them, which is likely caused by a smaller number of jobs. Tan et al. [17] proposed an approach based on PCA to predict processor and memory utilization in the commercial data center.

Datrois et al. [18] proposed a method that uses machine learning algorithms that predict the 24-h availability of resources at the host level. Their predicting method is based on the employment of quantile regression to ensure an elastic composition between the potential amount of resources to reclaim and to define unused resources. Hardware components like the central processor, memory, storage, and network metrics have been predicted to ensure exhaustive availability is guaranteed. Sekma et al. [19] have developed basic guidelines for developing an appropriate system for predicting the availability of CPUs for such computing infrastructures. They proposed a process for constructing a predictor that automatically checks the assumptions of vector autoregressive models in a time series. The authors performed three different past analyses, that is, the last hours, the same hours of the previous days, and the same weekly hours of the previous weeks. The suggested prediction model uses a multi-state prediction method to select a corresponding predictor.

## 2. Proposed Method

This paper aims to apply fuzzy logic theory to evaluate the CPU utilization by including the impact of RAM and storage utilizations to the CPU performance, while simultaneously running multiple applications processing. We used the system monitor (perfmon.exe) of the operating system to collect the performance data of hardware components, which are the processor time, swapping(pages), and disk time characteristics, accordingly CPU, RAM, and storage utilization information. During the monitoring, we built a dataset of RAM, storage and CPU utilizations from the test-bed computer. The view of the dataset is shown in Table 1.

**Table 1.** Database for training the Sugeno model.

|  | RAM Utilization (%) | Storage Utilization (%) | CPU Utilization (%) |
|---|---|---|---|
| 1 | 27 | 44 | 14 |
| 2 | 38 | 50 | 20 |
| 3 | 72 | 94 | 44 |
| … | … | … | … |
| 1000 | 85 | 100 | 66 |

A fuzzy expert system that uses human knowledge and their experience in this field to create linguistic descriptors for variables and to create fuzzy sets that allow the behavior of the studied phenomena to be controlled, or even as a tool for decision making [20]. In this work, we developed two types of FIS model (Sugeno and Mamdani) to evaluate the CPU utilization status by considering the influence of memory and storage utilization on CPU utilization. The difference between the FIS models are knowledge base and output, i.e., the Sugeno FIS uses a database and outputs quantity value of CPU utilization and the Mamdani FIS uses a rule base and performs linguistic value on the utilization state of CPU, like low, normal or high.

The architecture of the proposed fuzzy inference system is shown in Figure 1. The model contains fuzzification, knowledge base, inference engine, and defuzzification modules. The algorithm of the fuzzy inference system is described in Algorithm 1.

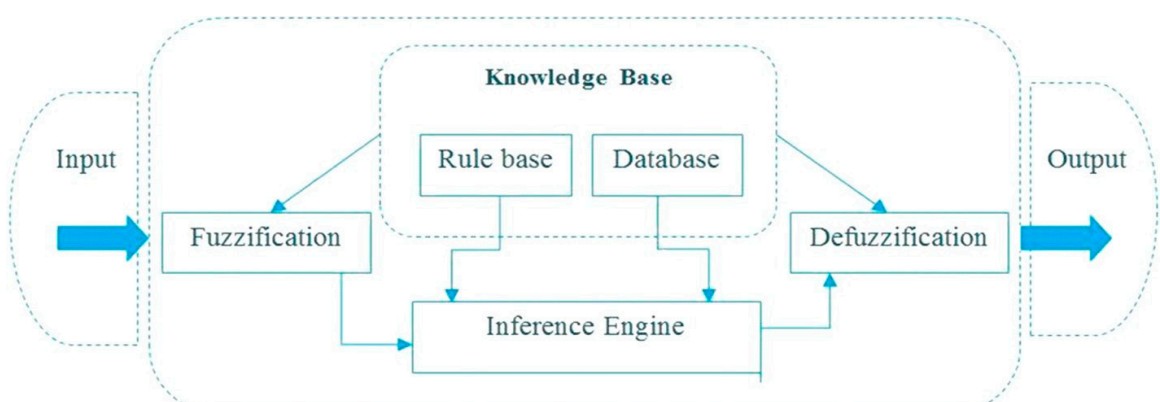

**Figure 1.** Flowchart of Fuzzy inference system.

---

**Algorithm 1.** Algorithm of Fuzzy inference system

---

1. Definition of linguistic variables
2. Construction membership functions
3. Construction knowledge base (rule base and database)
4. Fuzzification crisp input values
5. Training and evaluation knowledge base
6. Combining the results of each rule
7. Defuzzification non-fuzzy output values

---

The fuzzification process is the first step in the fuzzy inference system. The crisp input values of RAM and storage are converted to fuzzy inputs by a fuzzification module, which assigns the degree of membership to fuzzy sets defined for variables. We used the Gaussian membership function for the fuzzy set as shown in Figure 2 and Equation (1).

$$gaussian(x; c, \sigma) = e^{-\frac{1}{2}\left(\frac{x-c}{\sigma}\right)^2}$$

(1)

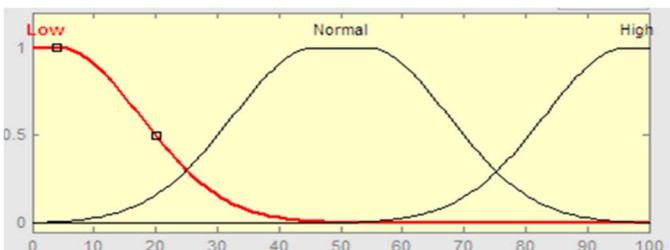

**Figure 2.** Gaussian membership function.

Equation (1) shows that c represents the center and $\sigma$ determines the width of the membership function. In Figure 2, the described membership function with the "normal" linguistic term defined by the function of Gaussian (x; 50, 25). This means that the center of membership function 50, where the function reaches the maximum value 1. Equation (1) uses all input and output variables to make membership functions.

A fuzzy inference engine can draw conclusions or predict the results of a system using a knowledge base. In the proposed FIS, the inference engine is based on a set of IF-THEN rules. As aforementioned, we developed two types of FIS. These are the Mamdani FIS and the Sugeno FIS. Our proposed Mamdani-type FIS is implemented by the following steps:

1.  Fuzzification input crisp values
2.  Built fuzzy rules
3.  Finding outcomes of the rules by combining the rule strength and output
4.  Get output distribution by combining the outcomes
5.  Defuzzification of the output

$$IF \ (RAM \ is \ Low \ and \ Storage \ is \ Low)THEN \ (CPU \ is \ Low) \tag{2}$$

$$IF(RAM \ is \ Low \ and \ Storage \ is \ Normal)THEN(CPU \ is \ Low) \tag{3}$$

The inference engine steps of the Sugeno model are almost the same as these steps, but the Sugeno FIS generates rules by a training process. Training occurs using the database, which is illustrated in Table 1. After training, the model builds optimal weights for prediction CPU utilization.

The detailed process of the Mamdani and Sugeno inference engine is illustrated in Figure 3. In Equations (2) and (3) we define two rules using two inputs (the full rule base is described in Figure 8). In these equations, RAM and storage are fuzzy values. Two inputs are fuzzified by applying an intersection on the crisp input values using a membership function. We used the "and" operator for combining two fuzzified inputs to obtain the rule strength. The Mamdani and Sugeno inference engine uses a membership function for each rule and then, according to the condition of the rule, reaches a conclusion.

In the last step, the defuzzification module converts the resulting fuzzy output, that is, the Sugeno FIS outputs crisp value and Mamdani FIS outputs linguistic value on CPU utilization. Here the scalar value of the fuzzy system is fuzzified, the rules are applied, the training model is built, and each rule generates fuzzy outputs and converts them to a scalar quantity. We used the centroid defuzzification method for the Mamdani FIS and the weighted average defuzzification method for the Sugeno FIS as shown in Figure 3.

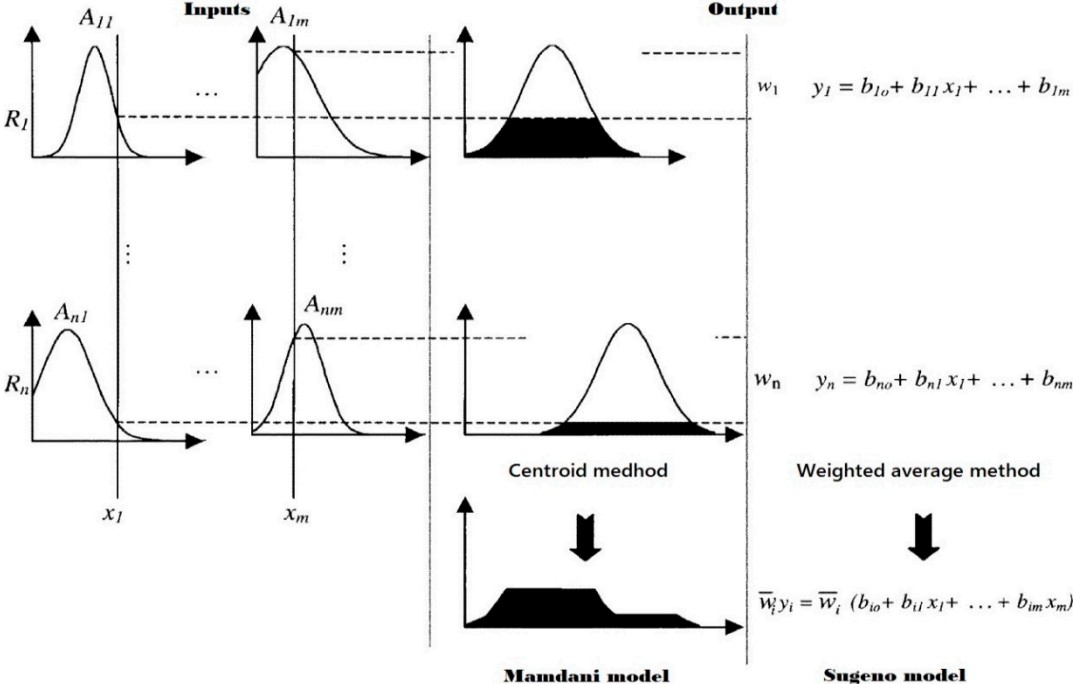

**Figure 3.** Two inputs and two rules based on the Mamdani and Sugeno inference engine.

The linguistic variables like low, normal and high are the criteria for evaluating the CPU, RAM, and the storage represented in the linguistic descriptors as the value ranges. For example, the variable CPU has a range of values from 0 to 40, which is considered "Low", from 20 to 80 is considered "Normal", and from 60 to 100 is considered "High". "Low", "Normal", and "High" are linguistic descriptors for the value sets for all variables of CPU, RAM, and storage.

As in principles of the fuzzy theory, sets of values overlap, and thus a value may partially belong to the set and have a membership degree of $0 \leq \mu \leq 1$, which can be any point between zero and one, where $\mu$ represents the degree of membership [21]. Consequently, the value belongs to several sets, and the total membership is added to one. The linguistic descriptors "Low" and "Normal" are two fuzzy sets of CPU variables that can overlap, so a CPU of 20 can be mostly "Low" with $\mu = 0.7$ and somewhat "Normal" with $\mu = 0.3$.

Thus, the fuzzy system converts the crisp input values into fuzzy inputs through a fuzzification block, which sets the degree of belonging to fuzzy sets for previously defined variables. The fuzzy system then uses a rule base developed by a human expert to predict a fuzzy inference of the phenomenon under investigation, as a result of which the fuzzy system has a defuzzification block that converts the fuzzy output to a crisp value. In this work, we built a fuzzy inference system with two crisp input variables and one output as shown in Figure 6.

## 3. Performance Evaluation

### 3.1. Experimental Environment

#### 3.1.1. Performance Monitor

The performance monitor package provides real-time system performance utilities. This program receives performance data from computer components. The performance monitor window allows us to view, in real-time, detailed information about the hardware components (CPU, disk, networks, and memory) and system resources (including descriptors and modules) used by the operating system, services and running applications. In addition, we can use the resource monitor to manage processes, start and stop services, analyze process deadlock, view thread wait chains, and determine the files

responsible for the blocking processes. When using the performance monitor, the CPU utilization is displayed in different fields. The "% User Time" shows the CPU utilization that occurred during execution at the application level. This field indicates the time taken to start the virtual processors. The "% Idle Time" indicates the percentage of time during which the CPU or CPUs were in idle mode, and the system did not have a failed request for disk I/O. The "% Processor Time" gives the summation of all the others and hence is the absolute CPU utilization as shown in Figure 4. In the "Memory" section, almost the same information is displayed that can be obtained with the help of the "Task Manager", but it is ordered in relation to the entire system. The histogram of physical memory displays the current organization of this memory, dividing the entire amount of memory into reserved, used, modified, in standby mode and free. Here "Pages/sec" shows memory utilization. To get disk utilization, monitoring the "Average transfer rate/sec", "Disk Queue Length", and "Disk Time (%)" pointers are more efficient pointers.

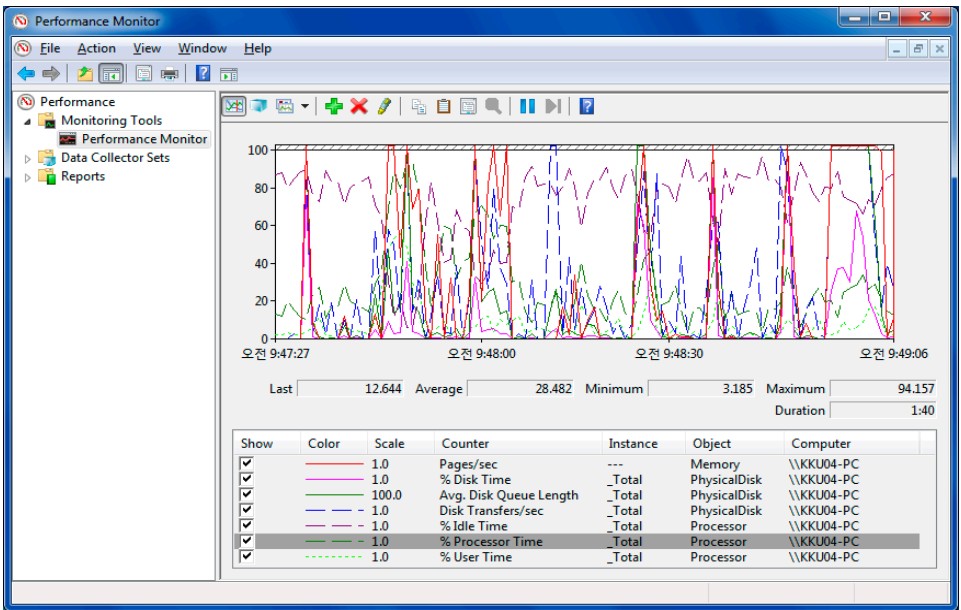

**Figure 4.** The main window of the performance monitor.

### 3.1.2. Fuzzy Logic Toolbox of MATLAB Environment

Fuzzy Logic Toolbox (FLT) is a MATLAB toolkit using the fuzzy logic, which provides analyzing, designing, and simulating systems of functions, applications, and a Simulink block. The fuzzy logic model contains the stages of designing fuzzy inference systems. Functions of the Fuzzy Logic Toolbox are provided for many common methods, including fuzzy clustering and adaptive neuro-fuzzy learning. The toolbox allows one to simulate complex system behavior using simple logical rules, and then implement these rules in a fuzzy inference system.

Fuzzy inference is a method that interprets values in the input vector and, based on user rules, assigns values to the output vector. Using editors and viewers in the Fuzzy Logic Toolbox, we can create a set of rules, determine membership function and analyze the behavior of the fuzzy inference system (FIS). We used all of the following editors and viewers to design our FIS models:

- FIS Editor—we used to create and select the type of fuzzy output system.
- Membership Editor—we edited the membership functions associated with FIS input and output variables.
- Rule Editor—Using this editor, we developed and edited the Mamdani FIS fuzzy rules.
- Rule Viewer—Shows detailed FIS behavior to help diagnose the behavior of specific rules or examine the impact of changing input variables.

- Surface Viewer–This generates a three-dimensional surface from two input variables and the FIS output.

### 3.1.3. Testbed Details

The Testbed details are shown in Table 2.

**Table 2.** Testbed parameters.

| Software and Hardware Components | Type and Capacity |
|---|---|
| Operating system | Windows 7 Enterprise, 64 bit |
| Hard-disk | 500 GB |
| Processor | 4 core, 3.10 GHz |
| RAM | 8.00 GB |

In our experimental testbed, we ran several applications simultaneously. In order to do performance analysis of CPU utilization, we monitored three hardware components such as processor time of CPU, disc time of storage, and paging of RAM. We considered that the utilization of RAM and storage are effects on CPU utilization.

### 3.2. Experimental Details

The CPU utilization evaluating process includes the following steps:

1. Fuzzification of input RAM, physical disk values
2. Determination rules of application and method of inference
3. Defuzzification CPU utilization values

Fuzzification of CPU utilization was performed using input variables and their membership functions of fuzzy sets. Results on the percentage of the utilization of RAM and storage, obtained from the input variables of the expert system based on fuzzy logic. Each input variable has three membership functions, that is, Gaussian membership functions were used in this proposed research paper. The fuzzy sets of input variables are listed in Table 3, and for both inputs (RAM and storage) the membership functions are shown in Figure 5.

**Table 3.** Fuzzy set of output variables.

| Linguistic Variable Name | Interval |
|---|---|
| Low | (0, 0, 25, 40) |
| Normal | (20, 50, 80) |
| High | (60, 80, 100, 100) |

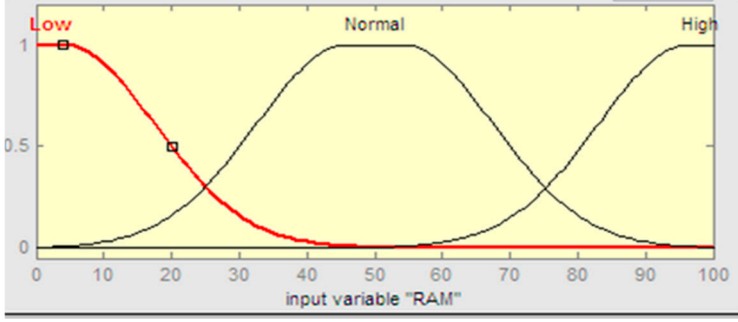

**Figure 5.** Membership function of RAM and storage.

RAM and storage are two input variables. The output variable is the CPU utilization value that is defined by fuzzy logic, as shown in Figure 6. FIS accepts crisp input data, fuzzifies them,

uses fuzzy on-premise operators (antecedent), implements inferences from the premise for output (sequential), combines the findings from fuzzy rules for creating a fuzzy inference and defuzzifies it to get a crisp conclusion. The proposed model in this paper is evaluated and tested using the Mamdani and Sugeno FIS.

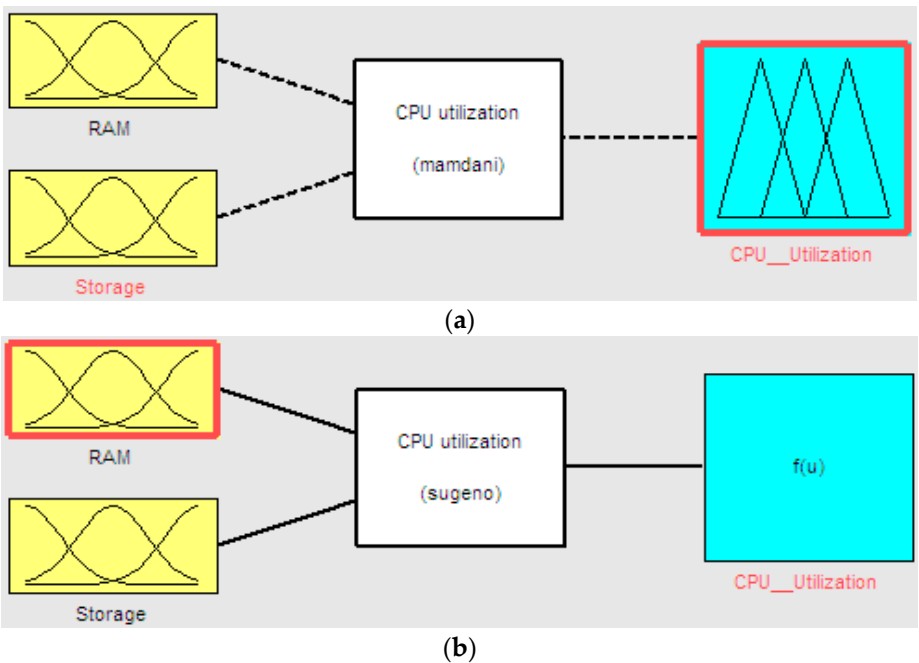

**Figure 6.** Input and output variables: (**a**) Mamdani FIS model; (**b**) Sugeno FIS model.

After defining inputs, the membership functions are constructed giving them ranges respectively. Membership functions are constructed to use in fuzzification and defuzzification. They map the non-fuzzy inputs to fuzzy values, which are then inferred according to the rule base. Figure 7 depicts the output variable with its member functions plotted on the curves. The three membership functions are mapped for each input data accordingly.

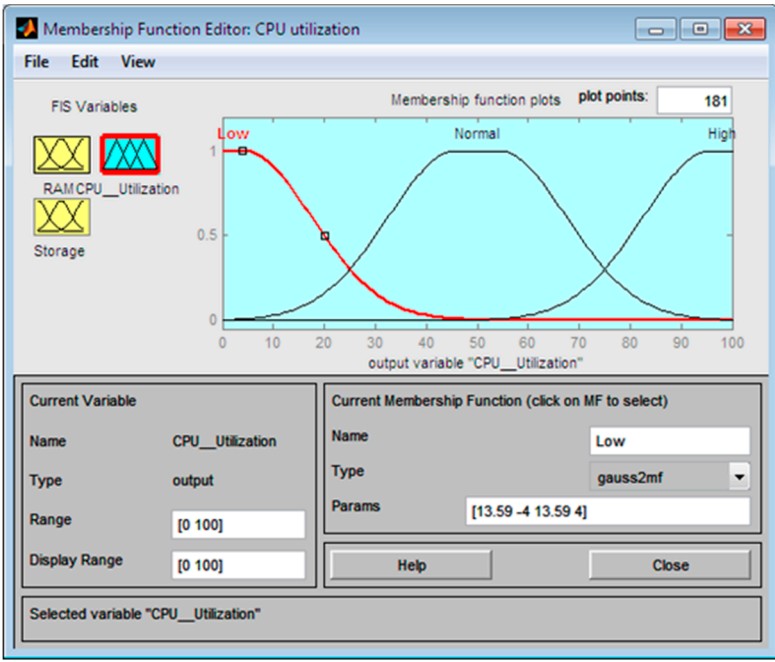

**Figure 7.** Construction of the output variable membership function of the Mamdani FIS model.

The fuzzy rules editor is used to define the rules for the parameters being tested. The rules define the input and output membership functions that will be used in the output process. These rules are linguistic and also have a "*IF-THEN*" form. Here the two parameters, RAM and storage, are linked through the *AND* logical operation. These fuzzy set logical operations are used to perform the rules evaluation and then combining the results of the individual rules. Figure 8a shows the work of the inference engine of the Mamdani FIS.

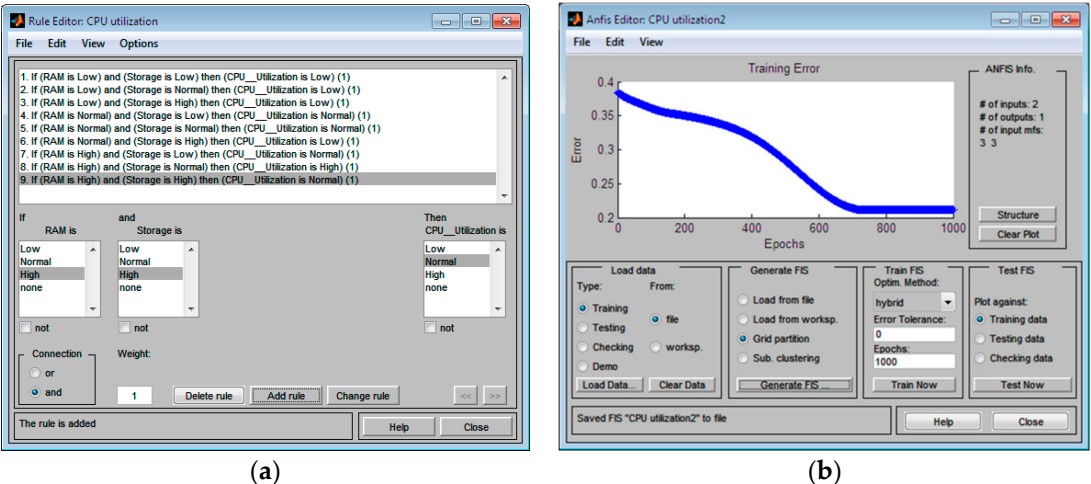

| (**a**) | (**b**) |

**Figure 8.** Defining rules for CPU utilization: (**a**) by rule base for the Mamdani FIS: (**b**) by training the dataset for the Sugeno FIS.

We defined nine rules for estimating CPU usage. In case several rules are active for the same output function of membership, it is necessary that only one value of membership is selected. This process is called "fuzzy decision" or "fuzzy inference". Several authors, including Takagi-Sugeno, Mamdami, and Zadeh developed a number of methods for fuzzy decision-making and fuzzy inference.

The Mamdani FIS rules are generated using linguistic variables for both the premise and conclusion. The Mamdani FIS Model has established relationships between variables to evaluate the FIS model by highlighting fuzzy rules in terms used by a human expert in the form of IF-THEN [22]. The formula used for the centroids defuzzification method in the FIS Mamdani is as follows:

$$Z = \frac{\int \mu_c(z) * z \, dz}{\int \mu_c(z) \, dz} \qquad (4)$$

The Sugeno FIS accepts a premise as a linguistic variable; however, its final part is a function, which can be zero (constant) or first order. The formula used by the Sugeno FIS for the weighted average defuzzification method is as follows [22]:

$$Z = \frac{\sum \mu_c z * z}{\sum \mu_c z} \qquad (5)$$

The weights are determined by the learning process. Training data are received from the system monitor of hardware components, which is illustrated in Table 1. The Anfis Editor window of the Fuzzy Logic Toolbox is designed to train the inference engine of the Sugeno FIS. The learning algorithm uses a hybrid algorithm, which is a combination of the gradient descent method and the least squares method. It is used to identify the parameters of fuzzy inference systems such as Sugeno. The learning process is shown in Figure 8b, training is completed in 1000 epochs and accuracy is equal to 0.211. The achieved accuracy of the training process is quite applicable for our Sugeno FIS.

When the three parameters were given different input values, i.e., low, medium and high, the overall result is a fuzzy value. The output is defuzzified after the output of the used rule base.

The membership function, previously defined, is used to defuzzify the output value. In Figure 9, the Mamdani FIS and Sugeno FIS input parameters are shown with RAM = 75.2 and Storage = 96.2. The CPU utilization of value 45.3 was evaluated by the Sugeno FIS. The Mamdani FIS rated CPU utilization of 41.9 in accordance with established rules. The cumulative result is shown by a red line in the graph showing the status "Normal". In the Rule Viewer and the Surface Viewer (Figure 10), the user can see the input changes and affect the output for the Rule Viewer and the Surface Viewer shown in three-dimensional graphics.

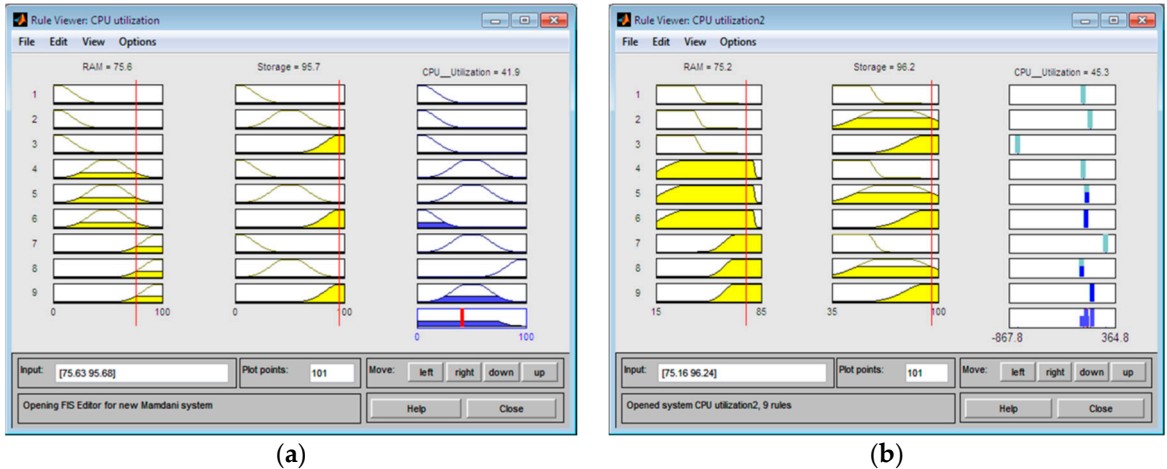

(**a**)　　　　　　　　　　　　　　　　　(**b**)

**Figure 9.** Rule Viewer showing the defuzzified form of parameters: (**a**) The Mamdani FIS model, the combined result is shown with the red line on the graph, showing the "Middle" status of CPU utilization; (**b**) The Sugeno FIS model outputs a numerical value.

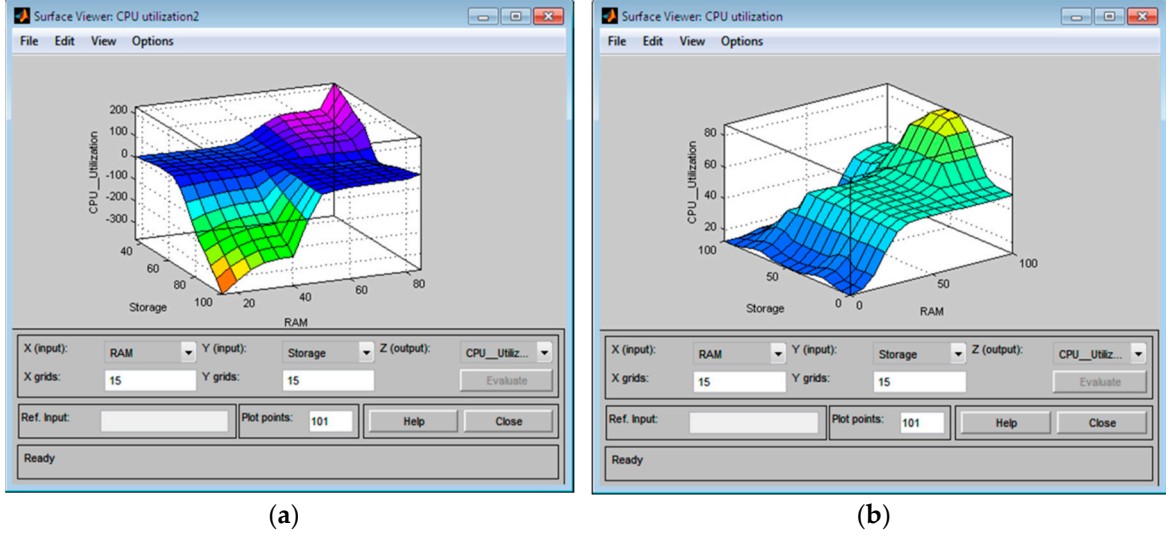

(**a**)　　　　　　　　　　　　　　　　　(**b**)

**Figure 10.** The surface viewer of CPU utilization: (**a**) The Mamdani FIS model; (**b**) the Sugeno FIS model.

*3.3. Experimental Results*

The CPU utilization is the number of processor cycles utilized by the process. In our experiment, we simultaneously launched a process of several applications to evaluate the effect of utilization RAM and storage to CPU utilization. In our paper, we analyzed RAM, storage and CPU, which can be updated, modified or configured by any user when some bottlenecks are detected during the system work. The cache has a huge influence on overall performance but we did not consider cache memory because it is an embedded system, which cannot be updated or modified. This section contains a simulation analysis of the proposed models. To demonstrate the performance of neuro-fuzzy logic

models (Mamdani FIS and Sugeno FIS), they were compared with the actual state of CPU utilization obtained from the performance monitoring program. The performance monitor gives only numerical real-time data about the state of each hardware component. Comparing the results of FIS models and the performance monitor gives the opportunity to evaluate the reliability of the FIS models.

　　A twenty-four-second duration state was monitored with various characteristics, such as processor time (CPU), disk time (storage) and paging (RAM). At the same time, acquired data was evaluated by the Mamdani and Sugeno FIS models. The FIS model evaluated the utilization of CPU and performed the results. In Table 4, the first three columns are data obtained via performance monitoring. The remaining columns represent the results of the numerical and linguistic value of the assessment of the Mamdani FIS and Sugeno FIS models. As we see, the numerical results of the Mamdani and Sugeno models is quite close to the actual state of CPU utilization. This means that our proposed Fuzzy inference systems are evaluating the true state of CPU, i.e., the built rule base and trained database sufficiently learned by inference engines. The linguistic results presented by the FIS models are more convenient and provide a demonstrative conclusion to the user about CPU utilization.

**Table 4.** Utilization information and FIS model results.

| RAM Utilization (%) | Storage Utilization (%) | CPU Utilization (%) | CPU Utilization by Mamdani FIS | | CPU Utilization by Sugeno FIS | |
|---|---|---|---|---|---|---|
| | | | Crisp Value (%) | Linguistic Value | Crisp Value (%) | Linguistic Value |
| 75 | 90 | 40 | 42.3 | Normal | 39.5 | Normal |
| 68 | 82 | 32 | 31.7 | Normal | 32.3 | Normal |
| 56 | 64 | 34 | 49.4 | Normal | 34.8 | Normal |
| 72 | 94 | 44 | 35.1 | Normal | 43.4 | Normal |
| 38 | 50 | 20 | 49 | Normal | 20.4 | Low |
| 56 | 72 | 32 | 44.7 | Normal | 33 | Normal |
| 30 | 44 | 14 | 47.2 | Normal | 14.1 | Low |
| 45 | 60 | 24 | 49.8 | Normal | 24.5 | Low |
| 40 | 60 | 25 | 48 | Normal | 24.4 | Low |
| 20 | 42 | 12 | 30 | Normal | 13.3 | Low |
| 50 | 80 | 34 | 32 | Normal | 34.8 | Normal |
| 30 | 50 | 20 | 47.2 | Normal | 20.1 | Low |
| 45 | 60 | 25 | 49 | Normal | 24.6 | Low |
| 68 | 90 | 40 | 28.1 | Normal | 40.1 | Normal |
| 55 | 74 | 34 | 43.2 | Normal | 33.5 | Normal |
| 40 | 60 | 24 | 50.2 | Normal | 24.4 | Low |
| 80 | 100 | 50 | 46 | Normal | 51.2 | Normal |
| 44 | 65 | 28 | 49 | Normal | 28.8 | Normal |
| 66 | 88 | 38 | 24 | Low | 38.2 | Normal |
| 75 | 96 | 44 | 42 | Normal | 45.3 | Normal |
| 85 | 100 | 66 | 50 | Normal | 65.9 | Normal |
| 60 | 79 | 32 | 32.6 | Normal | 31.8 | Normal |
| 60 | 88 | 38 | 18.6 | Low | 37.6 | Normal |
| 39 | 63 | 26 | 49.7 | Normal | 24.4 | Low |

　　To analyze the results, we made a graphical visualization. Figure 11a shows the effect of the RAM and storage utilization separately to the CPU utilization at the same time. The line with circles represents dependency on RAM utilization and CPU utilization. The line with squares represents storage and CPU dependency. We can see that the line graph shows that CPU utilization is increased accordingly by increasing RAM and storage utilization, i.e., utilization of these two objects increases or decreases CPU activity. In addition, the graph shows that RAM utilization more significantly affects it compared to storage utilization. Figure 11b presents twenty-four cases of the combined influence of RAM and storage on CPU utilization. The meaning of this graph is the same as analyzing Figure 11a. The graph shows that, in the tested computer, the regularity, influence, and congruence

between the hardware components are linear and each component is compatible with each other. The compatibility of the system means that there is no necessity for computer designing, and configuration enhancement [6]; that is, the system is managing with the given workload.

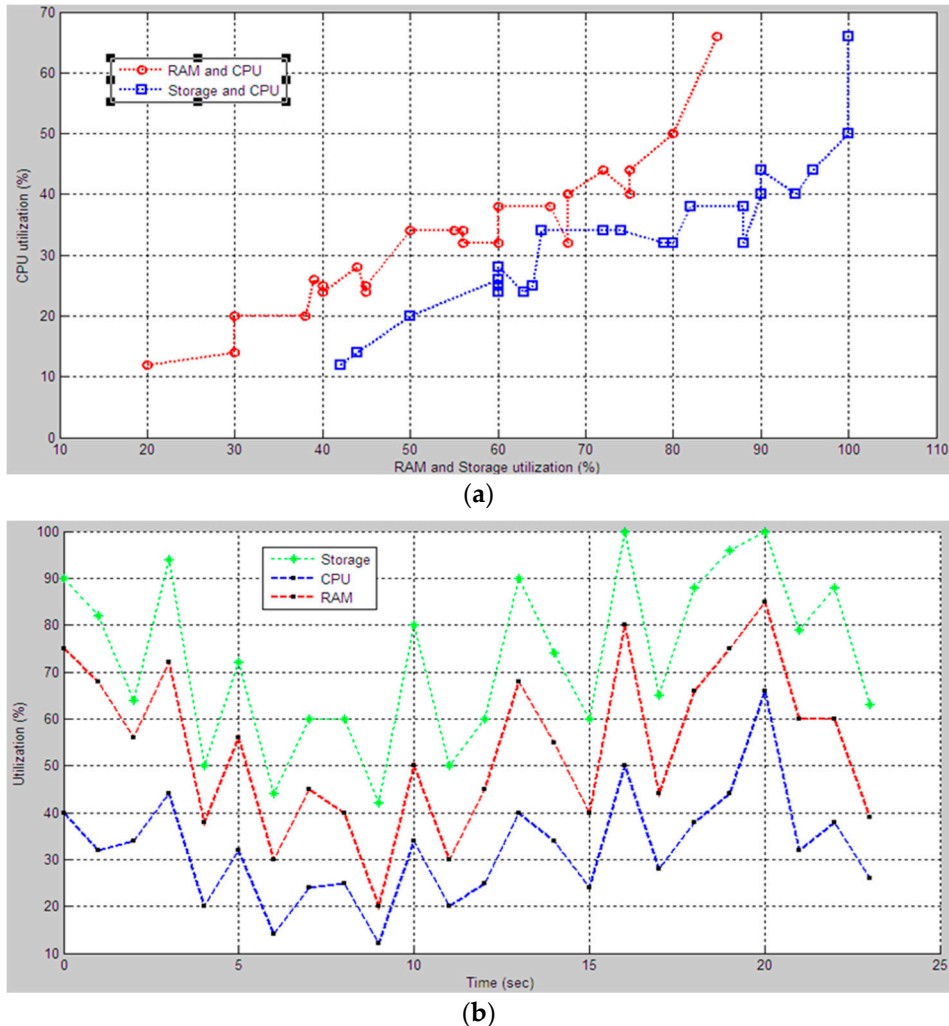

**Figure 11.** CPU utilization graph: (**a**) separate dependency of RAM and storage on CPU; (**b**) general dependency of RAM and storage on CPU.

The models' evaluation results, considering the influence of each factor and its index in the evaluation of the CPU utilization, are not the same. Figure 12 shows the CPU utilization graph for the output of the Mamdani FIS and Sugeno FIS. Figure 12a shows a comparison of the Mamdani FIS with CPU utilization, which is monitored through performance monitoring. The comparison shows that there are some unsuited cases with evaluation, but the results are close to each other. At the bottom of the figure, we show the difference between the real data and the Mamdani FIS result. The Sugeno FIS model shows excellent results, i.e., the model evaluation values are almost the same as the monitoring data (Figure 12b). The training of the inference engine was performed, rather than the rule base inference engine.

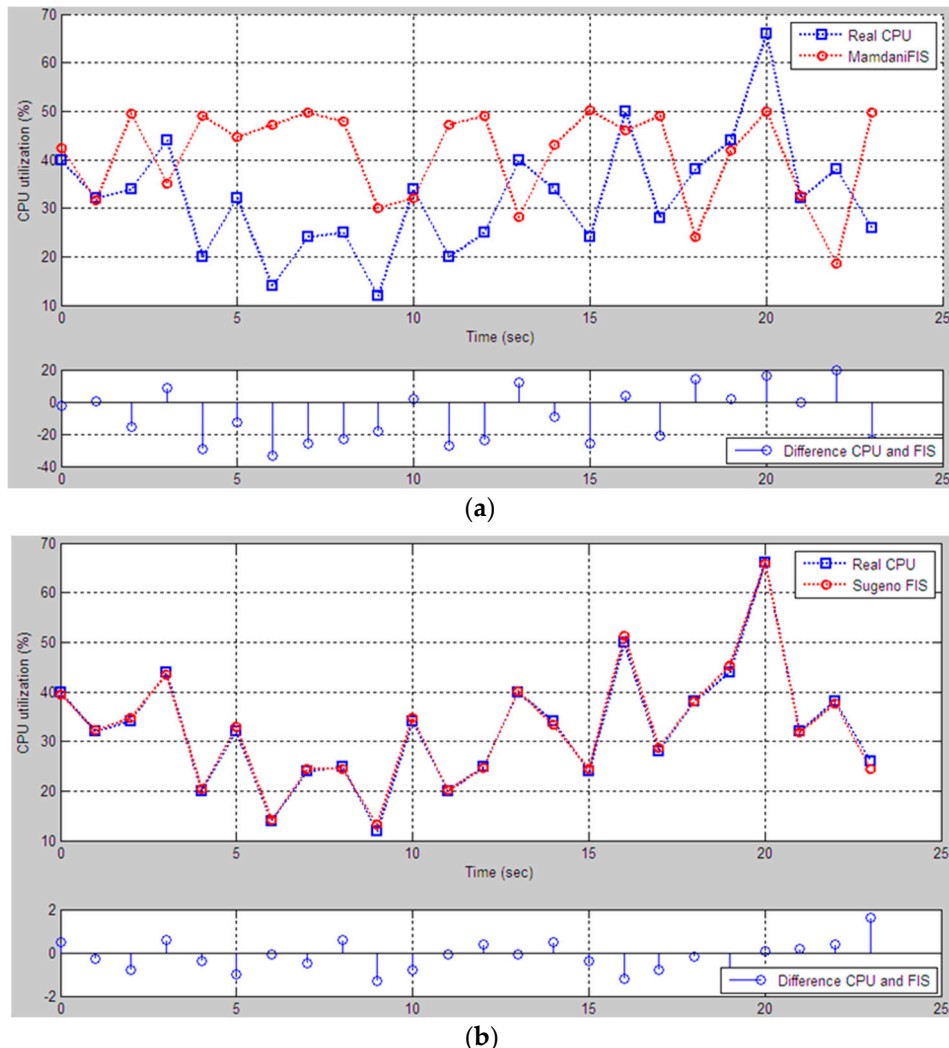

**Figure 12.** Comparison of results between the performance monitor and FIS models: (**a**) The Mamdani FIS; (**b**) the Sugeno FIS.

## 4. Discussion and Conclusions

The proposed fuzzy logic model for CPU performance evaluation is based on certain parameters that broadly represent the usual parameters like average workload percentage related to memory and storage. We have made the assumptions that the RAM and storage incorporate all the basic factors involved in computing system infrastructure, i.e., CPU load. The fuzzy logic system was used to define the rules for evaluation. An artificial neural network, some simulated environment or the real data values from the computing system were used to evaluate the performance. This fuzzy logic system is beneficial for those who are only learning about computer performance and how to evaluate hardware utilization. Computational experiments were conducted based on a neuro-fuzzy approach, to assess the impact of loading a physical disk and RAM on the processor state. The components of the input data are defined as term sets of the type low, middle, and high. The obtained results of computational experiments allow us to interpret the characteristics of computing systems in a linguistic form, i.e., in the natural language form, which is close to the judgment of specialists, which will significantly speed up the decision-making process. In this paper, the results are not compared with the result of benchmark experience data because the knowledge base was built in an individual testbed computer, i.e., the concept of our model requires an individual knowledge base. The generated rule base and data base cannot be used in the FIS model for evaluating the CPU utilization. We compared the results only with data that was received from the performance monitor. We did this comparison to

show the reliability of the FIS models. Because the performance monitor shows only numerical data about hardware utilizations, the program does not analyze the influence between components. Our FIS models analyze the receiving performance data and then evaluate the CPU utilization. If data coming from the performance monitor and the FIS results are close to each other, then the model is reliable and we can use it to predict the CPU utilization and to define the compatibility of components. The approach can be applied to personal computers, large mainframes, and supercomputers, and centralized and distributed systems using the own knowledge base of all these systems. The security issues and privacy concerns are assumed to be in a separate dimension; just the performance-affecting factors are involved in the logical modeling.

**Author Contributions:** Conceptualization, A.B.; Formal analysis, M.-C.K. and H.S.J.; Investigation, A.B.; Methodology, A.B., M.-C.K. and A.A.; Project administration, H.K.K. and A.A.; Resources, A.A. and H.S.J.; Software, A.B.; Supervision, H.S.J.; Validation, R.O.; Visualization, A.B.; Writing—original draft, A.B.; Writing—review & editing, H.K.K., R.O. and H.S.J.

**Funding:** This research was supported by Basic Science Research Program through the National Research Foundation of Korea (NRF) funded by the Ministry of Education (2018R1D1A1B07043417).

**Conflicts of Interest:** The authors declare no conflict of interest.

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
