# Peer review of "Application of Fuzzy Logic for Problems of Evaluating States of a Computing System"

_applsci, doi:10.3390/app9153021_

Round 1

Reviewer 1 Report

The authros have responded in a satisfactory manner to my main concerns regarding the experimental results (they have justified the lack of comparisons). The  references list has been updated with some newer articles.

In this paragraph -> In this paper, the result is not  compared with the result of benchmark experienced data. Because the knowledge base was built in individual testbed computer, i.e., the concept of our model requires individual knowledge base. The  generated rule base and data base cannot be used in FIS model for evaluating the CPU utilization. The approach can be applied to personal computers, large mainframes, and supercomputers, centralized and distributed systems using the own knowledge base of all these systems. 

please change result is to results are. Also, check again the paper for typos of this kind.

My only complain is that there is no reference to the cache memory, which is vital for the system's performance. I suggested this in my previous comments but i don't see anything added.

However, it is still an interesting paper worth of publication

Author Response

Response to Reviewer 1 Comments

The authors have responded in a satisfactory manner to my main concerns regarding the experimental results (they have justified the lack of comparisons). The references list has been updated with some newer articles.

Point 1: In this paragraph -> In this paper, the result is not compared with the result of benchmark experienced data. Because the knowledge base was built in individual testbed computer, i.e., the concept of our model requires individual knowledge base. The generated rule base and data base cannot be used in FIS model for evaluating the CPU utilization. The approach can be applied to personal computers, large mainframes, and supercomputers, centralized and distributed systems using the own knowledge base of all these systems. 

please change result is to results are. Also, check again the paper for types of this kind.

Response 1: We are agree. We have modified mentioned mistake in 485th. Also we have corrected this kind of mistakes in the whole text. Modified mistakes are highlighted.

Point 2: My only complain is that there is no reference to the cache memory, which is vital for the system's performance. I suggested this in my previous comments but I don't see anything added.

Response 2: We are fully agree with your opinion. Indeed, cache memory is vital for the system’s performance. In our work, we considered not embedded devices. It is mentioned in 416th line. In the future work, we are going to include cache memory. Now we cannot include cache due to data, during the experiences, we collected data only from the aforementioned components and now we do not know what the state of the cache was at that moment.

Reviewer 2 Report

It seems that Authors explain all key notions of proposed solution in clear and concise manner. A given description and information should be enough to a full reproduction of the proposed monitoring solution of computer system.

However, paper missed information about effectiveness of prototype solution (a discussion about others existing solutions was given in section 1.2.), and more precise, Authors should compare proposed system with others existing software (even one) for monitoring of computer system. In my opinion Authors show how their system works, but They do not show a real example even for small computer/workstation where proposed system will be used.  Obtained results should be compared with other available solutions and additional discussion should be shown the advantages of proposed system.

Author Response

Response to Reviewer 1 Comments

It seems that Authors explain all key notions of proposed solution in clear and concise manner. A given description and information should be enough to a full reproduction of the proposed monitoring solution of computer system.

Point 1: However, paper missed information about effectiveness of prototype solution (a discussion about others existing solutions was given in section 1.2.), and more precise, Authors should compare proposed system with others existing software (even one) for monitoring of computer system. In my opinion Authors show how their system works, but They do not show a real example even for small computer/workstation where proposed system will be used.  Obtained results should be compared with other available solutions and additional discussion should be shown the advantages of proposed system.

Response 1: We have experienced the purposed model in personal computer. The parameter of the personal computer is shown in Table 2. Obtained results are real experimental data. The results are compared with data received from Performance Monitor of Operating system. In the Section 4, we have added additional discussion. It is highlighted in 497th line.

This manuscript is a resubmission of an earlier submission. The following is a list of the peer review reports and author responses from that submission.

Round 1

Reviewer 1 Report

The results presented in the paper are impressive, but some details are needed to back up the value of them.

At a high level the paper lacks the motivation for what the work will be used for.  In the Introduction it jumps right into the meet and the conclusions do not tie things back up to a higher level.  Both would be helpful for giving the reader a better understanding of why the paper is important.

The largest issue with the paper is the lack of detail on the experiments run.  Its not clear if a training and test set were run.  The sizes of these sets and what applications were in them.  These experimental details are essential to understanding how flexible the model is and to verify its not overfitting to a small use case.

More interpretation of the end results would be helpful in explaining what is seen in the results.  Right now most of the data is presented without explaining what a reader is seeing and why it matters.  In particular, explaining outliers, such as, the first data point in figure 10b would be useful.

In general the writing could use some work.  A good proofreading pass to tighten this up would make reading much easier and improve the impact.

Reviewer 2 Report

Authors of this paper, initially provide a short survey on the basic concepts, characteristics, and parameters of computer systems that are responsible for the determination of the system performance, and the types of models that provide adequate modeling of these systems.

Then, they investigate and develop the applied aspects of the theory of fuzzy sets' principles and the Matlab environment tools for monitoring and evaluating the state of computing systems.

The subject of the paper is quite interesting and well presented. Some minor grammatical/syntax errors could be corrected during the preparation of the camera ready version.

Reviewer 3 Report

This work presents an interesting idea of applying fuzzy sets to monitor and evaluate a computer system's performance. However, this work needs improvement to be in a publishable form.

Specifically, the authors should consider the following issues:

Related work: The references include a lot of old and rather outdated works. Moreover, in the related work section, the authors describe some old papers but Kumar's work, which is rather new is described in one line. I think that the authors should extend this part, discuss the related work in more details and add newer papers. Also, some type of classification of the techniques presented should be included, so that the reader can understand the merit of this work.

Section 2 should include a full presentation. It is absolutely impossible to understand the idea based on one figure. Can you describe the fuzzification and defuzzification models? In line 170, the variables are the criteria.... but which variables? You mean the CPU, Memory and Storage. Then, how is the fuzzification model working?  The authors describe some correspondence between value range and description like high low, medium. How do these derive? The paper should be more stand-alone.

The results include no comparisons, but just the results coming from a performance monitor. How does this work compare to similar works? The authors should explain why there are no comparisons or add comparison results. Otherwise, the reader can't understand the benefits of this work. 

I am surprised why the cache memory is not included in the overall analysis. It is crucial in evaluating the performance of a system. The authors should at least add the cache memory in their work.